# MicroRNAs May Play an Important Role in Sexual Reversal Process of Chinese Soft-Shelled Turtle, *Pelodiscus sinensis*

**DOI:** 10.3390/genes12111696

**Published:** 2021-10-25

**Authors:** Tong Zhou, Hang Sha, Meng Chen, Guobin Chen, Guiwei Zou, Hongwei Liang

**Affiliations:** 1Yangtze River Fisheries Research Institute, Chinese Academy of Fisheries Science, Wuhan 430223, China; zhoutong@yfi.ac.cn (T.Z.); sh1812@yfi.ac.cn (H.S.); chenmeng@yfi.ac.cn (M.C.); CGB1251877642@126.com (G.C.); zougw@yfi.ac.cn (G.Z.); 2College of Fisheries and Life, Shanghai Ocean University, Shanghai 201306, China

**Keywords:** sex reversal, Chinese soft-shelled turtle, *Pelodiscus sinensis*, small RNA, microRNA

## Abstract

The Chinese soft-shelled (*Pelodiscus sinensis*) turtle exhibits obvious sex dimorphism, which leads to the higher economic and nutritional value of male individuals. Exogenous hormones can cause the transformation from male to female phenotype during gonadal differentiation. However, the molecular mechanism related to the sexual reversal process is unclear. In this study, we compared the difference between the small RNAs of male, female, and pseudo-female turtles by small RNA-seq to understand the sexual reversal process of Chinese soft-shelled turtles. A certain dose of estrogen can cause the transformation of Chinese soft-shelled turtles from male to female, which are called pseudo-female individuals. The result of small RNA-seq has revealed that the characteristics of pseudo-females are very similar to females, but are strikingly different from males. The number of the microRNAs (miRNAs) of male individuals was significantly less than the number of female individuals or pseudo-female individuals, while the expression level of miRNAs of male individuals were significantly higher than the other two types. Furthermore, we found 533 differentially expressed miRNAs, including 173 up-regulated miRNAs and 360 down-regulated miRNAs, in the process of transformation from male to female phenotype. Cluster analysis of the total 602 differential miRNAs among females, males, and pseudo-females showed that miRNAs played a crucial role during the sexual differentiation. Among these differential miRNAs, we found 12 miRNAs related to gonadal development and verified their expression by qPCR. The TR-qPCR results confirmed the differential expression of 6 of the 12 miRNAs: miR-26a-5p, miR-212-5p, miR-202-5p, miR-301a, miR-181b-3p and miR-96-5p were involved in sexual reversal process, which was consistent with the results of omics. Using these six miRNAs and some of their target genes, we constructed a network diagram related to gonadal development. We suggest that these miRNAs may play an important role in the process of effective sex reversal, which would contribute to the breeding of all male strains of Chinese soft-shelled turtles.

## 1. Introduction

The Chinese soft-shelled turtle, *Pelodiscus sinensis*, is an economically important aquatic species. The turtles are highly exploited in traditional Chinese medicine (TCM), but excessive wildlife exploitation may lead to biodiversity loss [1,2]. The development of artificial culture Chinese soft-shelled turtles may contribute to the conservation of biodiversity, which is the most important resource for humankind. For this to happen, it is necessary to prevent the escape/release of farm animals into the wild, thus preserving both *Pelodiscus* species and their distinct genetic lineages [3]. The male individuals of Chinese soft-shelled turtles have obvious advantages over female individuals in some growth characteristics, such as a larger body size, faster growth, a thicker and wider calipash, and less body fat [4], which is one of the most widespread phenomena in biology [5]. The tail of the male turtles exceeds the calipash, but the tail of the female does not, which is a common way to identify male and female Chinese soft-shelled turtles in production. A significant difference between male and female Chinese soft-shelled turtles is their exhibited sexual dimorphism [6]. Therefore, the male individuals have a higher economic value compared to the female individuals.

Sexual differentiation and differences in vertebrates have always attracted much attention [7]. The process of sex differentiation is regulated by multiple genes and mediated by related hormones, thereby inducing individual gonads to develop toward the testis or ovaries [8,9]. Many animals, including many aquatic animals, show obvious sexual dimorphism [10]. In some fish, such as Nile tilapia (*Oreochromis niloticus*), the males have a significantly faster growth rate than females, while female flounders (*Paralichthys olivaceus*) have a significantly larger size and growth than males. These obvious gender dimorphisms also provide new ideas for aquatic breeding.

The type of sex determination mechanism for the Chinese soft-shelled turtle is genotypic sex determination (GSD), not temperature-dependent sex determination (TSD), with the micro-sex chromosomes of the female heterogametic (ZZ/ZW) system [11]. The gonad differentiation of Chinese soft-shelled turtles begin at the 12th day and end at 22th day with 30 °C incubation [12]. Some gender-specific genes in Chinese soft-shelled turtles showed obvious effects in sex determination and gonadal differentiation, such as *Dmrt1, Sox9, Cyp19a*, and other sex-related genes [13,14].

The exogenous hormone estradiol can lead to the sex reversion of the male individuals of Chinese soft-shelled turtles into a pseudo-female individual (ΔZZ) with a female phenotype and a male genotype [14,15]. After the pseudo-female is sexually mature, it can be crossed with the male (ZZ) to obtain all male offspring. Therefore, the research of male to female sex reversal is the basis of all male breeding of Chinese soft-shelled turtles, and it is also important to understand the molecular mechanism of this process. However, the molecular mechanism of the sexual transformation of the Chinese soft-shelled turtle is still unclear.

Small RNAs play a significant role in various steps of fertilization, sex differentiation, gametogenesis, and embryogenesis in mammalian, especially microRNAs (miRNAs) [16,17,18,19]. MiRNAs are kind of small, highly conserved, non-coding RNAs, 18–26 nucleotides in length, and involved in the post-transcriptional regulation of the related genes [20,21,22]. Studies have shown that miRNAs regulate one-third of human genes [23] and abnormal miRNA profiles regulate tumor phenotypes through inhibiting their target genes [24]. Mechanisms for the miRNA-mediated downregulation of gene expression involve some combination of translational repression, mRNA deadenylation, decapping, 5′-to-3′ mRNA degradation, and alteration of mRNA stability [25,26].

MiRNAs as novel and highly conserved small RNAs play a vital role in sex determination and gonadal differentiation [27,28]. In the present research, various miRNAs have an effect during different developmental stages of gonads in teleosts [29]. With regard to gametes, let-7a-1-5p, let-7c-5p and miR-92b-3p expression was higher in spermatozoa, and miR-21-5p and miR-430b-3p expression was higher in oocytes [30]. MiR-141 and miR-429 have been implicated in testicular development and spermatogenesis [31]. However, the molecular mechanism of miRNAs on gonadal differentiation in Chinese soft-shelled turtles remains unclear.

The aim of the present study was to explore the relationship between small RNAs and males, females, and pseudo-females, to provide a new insight for the sex differentiation of Chinese soft-shelled turtles. In addition, investigating the role of miRNAs in Chinese soft-shelled turtles contributes towards improvement in breeding techniques for the holistic development of the aquaculture sector.

## 2. Materials and Methods

### 2.1. Maintenance of Chinese Soft-Shelled Turtles

Chinese soft-shelled turtles were provided by Anhui Xijia Agricultural Development Co. Ltd. (Bengbu, China). The fertilized eggs of P. sinensis were incubated in an egg incubator at 30 ± 0.5 °C, with the humidity maintained at 80–85%. The animal pole of eggs should point up and keep the humidity during hatch period, which lasts about 45 days [3]. The newly hatched turtles were kept in a 30 ± 0.5 °C greenhouse with commercial feeds (Jinjia, China) three times a day; the stocking density of juvenile turtles is about 100/m^2^.

### 2.2. Estradiol Treatment of Chinese Soft-Shelled Turtles

The fertilized eggs were incubated in the constant temperature humidity incubator at 30 ± 0.5 °C and 80–85% humidity for 15 days. We diluted estradiol (E2) with ethanol into a reagent of 10 mg/mL, dipped a cotton swab into a small amount of hydrochloric acid (HCl), and gently smeared it on the soft-shelled turtle fertilized eggs to make the eggshell soft. A micro-syringe was used to inject 5 µL of 10 mg/mL E2 into the soft-shelled turtle fertilized eggs. After 1 year of cultivation, the biological sex of the Chinese soft-shelled turtle was detected by the phenotype, and the genetic sex was analyzed by PCR with sex-specific primers (4085-f/r, col-f/r) (Table 1) [4].

### 2.3. Sample Collection and RNA Extraction

According to different genders, the differentially treated animals were divided into three groups (each group contained 6 animals): males, females, and pseudo-females. In order to obtain more differential miRNAs to investigate the molecular basis of sex reversal, the hypothalamus pituitary–gonad axis (HPGA) and brain pituitary–gonad axis were considered. Then, 5 tissues (i.e., heart, liver, muscle, gonad, and brain) were collected for RNA extraction. Total RNAs for subsequent analysis were extracted with the TRIzol reagent (15,596,026, Invitrogen, shanghai, China). RNA degradation and contamination was monitored on 1% agarose gels. RNA purity was checked using the NanoPhotometer^®^ spectrophotometer (IMPLEN, San Diego, CA, USA). RQ1 RNase-Free DNase (M6101, Promega, Madison, WI, USA) was used to remove the DNA of RNA samples and stopped the reaction with stop solution at 65 °C for 10 min. rRNA accounts for about 82% of the total RNA, but it provides very little biological information, therefore it needs to be removed before sequencing. The rRNA is removed by non-denaturing PAGE gel, which can separate nucleic acid sequences of different lengths. The length of rRNA is much larger than small RNA. Therefore, rRNA can be separated by recovering 18–40 bp fragments. The RNAs of 5 tissues for each sample were pooled together for small RNA sequencing analysis.

### 2.4. Library Preparation for Small RNA

A total of 3 μg RNA per sample was used as input material. Small RNA library was performed using NEBNext^®^ Multiplex Small RNA Library Prep Set for Illumina^®^ (NEB, New York, NY, USA) following manufacturer’s recommendations, and index codes were added to attribute sequences to each sample. Briefly, RNA adapters were ligated to 3′ and 5′ ends of RNA followed by cDNA synthesis and PCR amplification. The cDNA library was size-separated using PAGE gel, and small RNA between 18 bp and 40 bp was excised and purified.

### 2.5. Bioinformatic Analysis

Raw data in fasta format were processed through in-house perl scripts. In this step, clean data were obtained by removing reads containing adapter, reads on containing ploy-N, and low quality reads (reads having >50%, bases with quality score ≤5) from raw data using PRINSEQ (version 0.19.3) [32]. At the same time, Q20, Q30, and GC content of the clean data were calculated. In second-generation sequencing, each base measured gave a corresponding quality value (Q), which is a measure of sequencing accuracy. The higher the quality value (Q), the lower the probability (P) of the base being tested incorrectly. The calculation formula is Q = −10 lgP. Q20 and Q30 represent the percentage of a certain base quality value to the total number of bases. When the value of Q30 is higher than 85%, it indicates that the quality of sequencing is very good, and the next step of analysis can be carried out. Meanwhile, GC content is also an indicator of data quality, generally between 50 and 60%. All downstream analyses were based on the high-quality cleaned data.

To identify known miRNAs, we mapped sequenced reads to the sequences collected in mirBase using mirdeep2 [33]. We predicted candidate novel miRNA using mirEvo [34] and miRdeep [35] and assessed the length distribution and nucleotide proportion. The known miRNAs and novel miRNAs were combined as the final miRNA set.

The nucleic acid sequences of all genes identified by small RNA-seq in the database of turtle genome (AGCU00000000.1) [36] were used to blast against the Swiss-Prot and TrEMBL (the Swiss Institute of Bioinformatics and the European Bioinformatics Institute) protein database to get the UniProt-accession [37]. After obtaining the UniProt-accession of the genes, their KEGG Orthology ID and GO Orthology ID were obtained with the online tool bioDBnet (http://biodbnet.abcc.ncifcrf.gov accessed on 15 July 2021) for enrichment analysis.

### 2.6. Quantitative Real-Time PCR (qPCR)

The qPCR assays were performed to validate the small RNA-seq data. MiRNAs 1st Strand cDNA Synthesis Kit (by stem-loop) is a special kit suitable for one-strand synthesis of miRNA cDNA by stem-loop method which contains genomic DNA removal steps. Quickly remove the contamination of genomic DNA under the condition of 42 °C for 2 min to ensure that the follow-up results are more reliable (Vazyme, China). The subsequent quantification of cDNA products was performed as described previously [38]. U6 rRNA expression was used to normalize miRNA expression [39]. The primers are designed for this experiment by miRNA design. The primers for target gene qRT-PCR were provided in Table 1. Relative miRNA and mRNA expression levels were calculated by the 2^−(ΔΔCt)^ method [40].

### 2.7. Analysis of Differentially Expressed microRNAs and Target Genes

MiRNAs with a fold change ≥1.5 and a q-value ≤0.05 were considered as differentially expressed miRNAs. Differentially expressed microRNAs were classified with Venn’s diagrams by online tools (https://bioinfogp.cnb.csic.es/tools/venny/index.html accessed on 18 September 2021). Use MiRanda and qTar software to predict target genes for known miRNAs and new miRNAs. In order to ensure the accuracy of the results, the final result is the intersection of the two softwares [41]. Cytoscape tool was used to form a network diagram of miRNAs and target genes [42].

### 2.8. Statistical Analysis

All the experimental data from at least three independent experiments were analyzed using GraphPad Prism 7.0 software (San Diego, CA, USA) and were expressed as the mean ± SD. Student’s *t*-test were performed to compare the differences between two groups.

## 3. Results

### 3.1. Pseudo-Female Chinese Soft-Shelled Turtles Were Obtained by E2 Treatment

Exactly 5 µL of 10 mg/mL estradiol (E2) was injected into the soft-shelled turtle eggs using a micro-syringe, which were hatched at 30 ± 0.5 °C for 15 days (Appendix A) to induce the pseudo-female turtles. Whether the tail length exceeds the calipash is a common way to distinguish between male and female turtles in seed selection and breeding. After 1 year of cultivation, the results showed that the tails of some turtles treated by E2 could not exceed the calipash (Figure 1A), while PCR revealed they were male turtles by the sex-specific primers (Figure 1B, Table 1). Therefore, the Chinese soft-shelled turtles which own the feature of the male genotype and female phenotype were named pseudo-female turtles (Z).

Figure 1A showed that the tail length of pseudo-female turtles (Z) and female turtles (F) were similar, but obviously shorter than the tail length of male turtles (M). We conducted hormone induction on 219 soft-shelled turtles and tested their genotypes and phenotypes after 1 year. We found that the rate of sexually reversed soft-shelled turtles was 50.83% (Appendix A). These results showed that exogenous hormones E2 could cause sexual reversal in male individuals and generate pseudo-female individuals.

### 3.2. The Distribution and Number of Small RNA among Males, Females and Pseudo-Females

The exported data of omics showed that the raw reads of sample F, M, and Z were 27,912,069, 24,060,307, and 31,682,467. The Q20 and Q30 of the three groups were all above 99%; meanwhile, the GC contents were 49% (F), 51% (M), and 50% (Z), respectively (Table 2). The number of total reads and bases revealed that the male turtles had far less than the female turtles and pseudo-female turtles, which proved that small RNAs had a significant impact during sexual dimorphism (Table 3).

Besides the differences in the number of small RNAs, greater diversity was found in chromosome distribution. The distribution of group M was wider than the other two groups in the *P. sinensis* reference genome, which showed that some small RNAs are silenced during the transition from male to female by E2 treatment (Figure 2A). The sequence length distribution of three groups showed that the male group had more short fragments. In the groups of the females and pseudo-females, the 22 nt length transcripts were the most abundant, while the male group 18 nt, 19 nt, 20 nt, and 22 nt length transcripts all occupying a large part (Figure 2B). About 62.12%, 63.54%, and 72.60% of high-quality reads were mapped to the turtle genome (AGCU00000000.1).

The number of small RNAs in the three groups were 21,538,085, 13,550,606, and 20,669,068, respectively (Figure 2C). The distribution of small RNAs showed that the number of miRNAs in male turtles was strikingly less than the one of the pseudo-female turtles, which suggested that the miRNAs played an important role in the process of gonadal differentiation. Appendix A-C showed the comparison and annotation situation of unique small RNAs. The number of unique known miRNAs of males, females, and pseudo-females (F, M, Z) are 1653, 1090, 1517. The total amount of rRNA in the classification annotation results can be used as a quality control standard for a sample, which ensured that the three RNA samples met quality standards (Appendix A).

### 3.3. The Character of miRNAs Identified by Males, Females and Pseudo-Females

MiRNAs as a pivotal component of small RNAs showed significant difference among the males, females, and pseudo-females. The correlation of miRNA expression levels between samples is a key indicator. The closer the correlation coefficient is to 1, the higher the similarity of the expression patterns between samples. Compared with male individuals, the correlation showed that the pseudo-female individuals are more similar to that of female individuals (Figure 3A), and showed that the miRNAs have changed from males to females in the process of sexual reversal.

The miRNAs of male individuals are significantly different from the female and pseudo-female individuals according to the boxplot and distribution of miRNA TPM density in the different samples (Figure 3B and Appendix A). The process in which the miRNAs develop from a precursor to a mature body is completed by Dicer digestion, the specificity of the restriction site gives the first base of the mature miRNA sequences a strong bias. Therefore, we compared the first base of small RNAs and the base distribution of each position of mature miRNAs. Figure 3C showed that all of the first nucleotide of female and pseudo-female individuals was U when the length reached up to 27 nucleotides. However, the male individuals lacked the miRNAs which were more than 27 nucleotides in length (Figure 3C). The difference in miRNAs among the three groups identified the prominent role of miRNAs in the process of gonadal differentiation. In another aspect, each nucleotide bias position of the miRNAs in males, females, and pseudo-females had no obvious difference (Appendix A).

### 3.4. Differential miRNAs Analysis of Males, Females and Pseudo-Females (F, M and Z)

According to the miRNA expression profiles, we analyzed and compared the differential miRNA for different groups. Figure 4A showed that there are 79 up-regulated miRNAs and 344 down-regulated miRNAs between female and male individuals. Figure 4B,C showed that pseudo-female individuals have 145 up-regulated miRNAs and 59 down-regulated miRNAs against female individuals, and 173 up-regulated miRNAs and 360 down-regulated miRNAs against male individuals (*p* < 0.5). Among the differential miRNAs, the number of differential miRNAs between males and pseudo-females significantly increased.

Then, we performed GO and KEGG analysis of the target genes of the differential miRNAs among the three comparison groups (Figure 4D,E and Appendix A). In the process of reversing from a male to female soft-shelled turtle, GO enrichment results showed that the target genes were mainly enriched in biological processes including cellular process, single-organism process, and biological regulation. Candidate target genes of differential miRNAs between males and pseudo-females were overrepresented in molecular function such as binding, catalytic activity, and signal transducer activity (Figure 4D). Meanwhile, the KEGG analysis of differential target genes between male and pseudo-female individuals showed that the main processes are transport and catabolism, signal transduction, and the immune and endocrine system (Figure 4E).

GO and KEGG enrichment analysis were performed on the differential target genes of the other two pairs. GO enrichment analysis showed that the target genes of differential miRNAs had a big impact on various processes such as the cellular process, single-organism process, and biological regulation (Appendix A). KEGG enrichment of the targets of female-biased miRNAs displayed overrepresentation of many important signaling pathways such as signal transduction and lipid metabolism (Appendix A).

### 3.5. Cluster Analysis of Differential miRNAs and Validation of the Target Genes

As shown in Figure 5A, we performed Wayne analysis on unique miRNAs identified in three groups. There are 278 male-specific miRNAs and 32 female-specific miRNAs. Different from male and female, pseudo-female individuals have 113 novel miRNAs (Figure 5A). A total of 737 miRNAs were significantly differentially expressed among the three groups (*p* < 0.5). The cluster map of miRNA expression patterns reflects the level of gene expression and expression patterns (Figure 5B). From the cluster map, we can clearly see that there was a striking difference in the expression of miRNAs between males and the other types. The expression levels of miRNAs for females and pseudo-females were significantly lower than the expression level of males.

From the differential miRNAs, we found 12 miRNAs related to gonad development and performed clustering heat map analysis (Figure 5C, Table 4). The number in Table 4 represents the expression level of some differential miRNAs related to gonadal development. The miRNAs related to gonad development included miR-143, miR-26a-5p, miR-22a-3p, miR-212-5p, miR-181b-3-3p, miR-212, miR-202-3p, miR-301a, miR-202-5p, miR-25-3p, miR-96-5p, miR-181b-3p. To validate the sequencing data of miRNAs, we selected six miRNAs to test their relative expression of three samples (Figure 6A–F). The expression of six miRNAs in the samples was consistent with the results of RNA sequencing. Among the miRNAs, miR-26a-5p, miR-212-5p, miR-301a, miR-202-5p, miR-181b-3p were up-regulated during the sexual reversal process, while miR-96-5p was down-regulated. These results suggest that miRNAs may play an important role in the process of sexual reversal.

### 3.6. Construction of the Gonadal Related miRNAs and Some Target Genes Network

The result of qRCR and transcriptome data revealed that miR-26a-5p, miR-212-5p, miR-301a, miR-202-5p, miR-181b-3p, and miR-96-5p were gonadal-related miRNAs. To construct the network of miRNAs and target genes, we screened some target genes which showed a strong correlation with gonadal development (Figure 7). The network diagram showed that the miR-26a-5p played a significant role in these processes, such as ovarian development, osteoclast-stimulating, osteoclast-associated, and growth/differentiation. MiR-212-5p and miR-202-5p were involved in the development of ovaries or testes. During the oocyte maturation process, the miR-301a played an important role. These miRNAs and some target genes may participate in regulating sexual reversal processes, including ovarian development and growth/differentiation.

## 4. Discussion

Sexually dimorphic phenotypes exist in many animals and have complex genetic architectures [10,43,44]. Chinese soft-shelled turtles have obvious sexual dimorphism. The advantage of male individual Chinese soft-shelled turtles, which exhibit a larger body size, faster growth, a thicker and wider calipash, and less body fat, gives male Chinese soft-shelled turtle individuals a high economic value [45,46]. Therefore, cultivation of the all-male phenotype of Chinese soft-shelled turtle is a hot issue in actual production.

In the previous research, Ma et al. and Liu et al. found that miRNAs and lncRNA were involved in gonadal development [47,48]. The discovery of small RNAs such as miRNAs, snRNAs, and other types had unveiled a slew of powerful regulators of gene expression in recent years [49]. The defining features of small ncRNAs are their short length (~20–30 nt), their association with members of the Ago (argonaute) family of proteins, and typically their effect on the downregulation/silencing of target gene expression [50]. Of these small RNA classes, microRNAs (miRNAs) have emerged as key regulators of biological processes in animals [51]. MiRNAs played a crucial role in gonadal differentiation. MiRNA-mediated steroidogenesis can regulate adrenal and gonadal steroidogenesis [52]. Liu et al. found 665 differentially expressed miRNAs, with 519 being up-regulated in testis and 146 being down-regulated in ovary, respectively [48]. A total of 633 miRNAs showed differential expression in the ovaries and testes [47]. However, the molecular mechanism of miRNA regulating gonadal differentiation is currently unclear. We identified 21,538,085, 13,550,606, and 20,669,068 small RNAs of males, females and pseudo-females, respectively, by small RNA-seq, and found 351 miRNAs related to sex, which greatly enriched the miRNA database related to gonadal differentiation.

Figure 2C showed the number of small RNAs of male individuals that was obviously decrease compared to the female and pseudo-female individuals. Analyzing the distribution of small RNAs with males, females, and pseudo-females, it was found that most of the small RNAs lacking are miRNAs. MiRNAs generally base-pair to mRNAs with nearly perfect complementarity and trigger endonucleolytic mRNA cleavage by the RNA interference (RNAi) mechanism [25]. We concluded that the number of miRNAs of pseudo-female individuals changed during the transition from male to female, the newly produced miRNAs may have a certain inhibitory effect on androgens. After comparison, it was found that the small RNAs of pseudo-female individuals induced by E2 showed similar characteristics to female individuals, while they were different from male individuals. The results of small RNAs and miRNAs showed that the similar features had changed although the genotype had not changed.

We found that the expression levels of miRNA in females and pseudo-females were significantly lower than that of males after clustering analysis of different miRNAs. Bhat et al. summarized the current advances made in teleost miRNA study regarding reproduction and gonadal development [29]. According to the previous study, 12 miRNAs, such as miR-143, miR-26a-5p, miR-22a-3p, miR-212-5p, miR-181b-3-3p, miR-212, miR-202-3p, miR-301a, miR-202-5p, miR-25-3p, miR-96-5p, and miR-181b-3p also exist in the gonadal development of Chinese soft-shelled turtles though small RNA-seq [30,53,54]. In this study, miRNAs such as miR-26a-5p, miR-212-5p, miR-301a, miR-202-5p, and miR-181b-3p were up-regulated during the sexual reversal process, while miR-96-5p was down-regulated. The network diagram showed the relationship of some miRNAs and target genes related to gonadal differentiation. MiRNAs are a class of endogenous small noncoding RNAs that participate in most biological processes via regulating target gene expression [55]. Multiple miRNAs or genes were not merely a single gene involved in regulating the biological processes. Wang et al. proved the inhibitory effect of miR-215-5p on recombinant PCDH9 containing its own promoter and 3′ UTR more effective than that containing the promoter or 3′ UTR alone, which indicated the benefit of synergetic suppression by miRNAs [56]. The high expression level of miR-26a-5p in male individuals may inhibit the expression of target genes such as growth/differentiation, ovarian development, and the oocyte division biological process. Meanwhile, the low expression level of miR-96-5p in male individuals may contribute to the production of estrogen. Currently, little is known about their functions in the gonad development of Chinese soft-shelled turtles. Given further study, we would analyze their function of sex-biased miRNAs to identify their role in depth. These sex-biased miRNAs including male-biased miRNAs and female-biased miRNAs were predicted as negative regulators of multiple genes that significantly upregulated or downregulated in turtle ovaries at maturation and spawning stage. Potential miRNA-mediated post-transcriptional regulation in gonads or testis is a clue to the gonad differentiation mechanisms of turtles. Taken together, these data suggested that miRNAs played a crucial role in the sexual reversal process from males to females. The characteristics the pseudo-female displayed were obviously close to females but away from males. During the sexual reversal process, small RNAs, especially miRNAs, affected the expression of some target genes. The quantitative results also identified that miRNAs played an important role in the sexual reversal process.

## 5. Conclusions

In this study, the pseudo-females were obtained by injecting estradiol, in which the sex was reversed with the male genotype and female phenotype. A total of 533 differentially expressed miRNAs were found in the process of transformation from male to female phenotypes, including 173 up-regulated miRNAs and 360 down-regulated miRNAs. Six miRNAs, such as miR-26a-5p, miR-212-5p, miR-301a, miR-202-5p, miR-181b-3p and, miR-96-5p, were identified as having a close relationship with sexual reversal process. Our results suggest that they might serve as important candidates for further investigation with regard to the biological functions of miRNAs in the sexual dimorphism of Chinese soft-shelled turtles. At the same time, it also provides a direction for the study of the molecular mechanism of sexual reversal.

## Figures and Tables

**Figure 1 genes-12-01696-f001:**
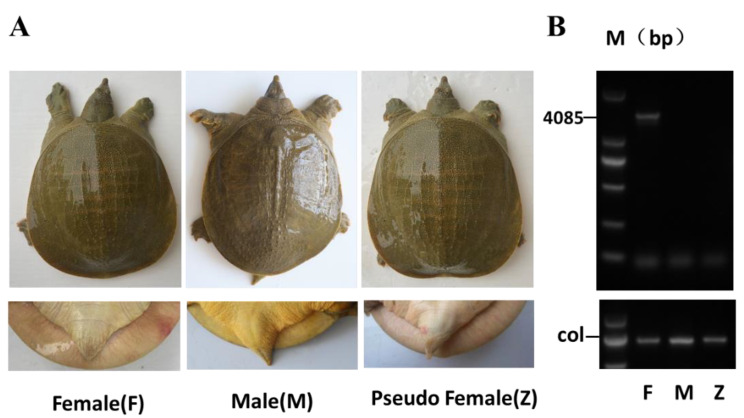
Pseudo-female Chinese soft-shelled turtles were obtained by E2 treatment: (**A**) Pseudo-female of Chinese soft-shelled turtles were obtained by E2 treatment. (**B**) Verification of three phenotypes by PCR. F: female, M: male, Z: pseudo-female.

**Figure 2 genes-12-01696-f002:**
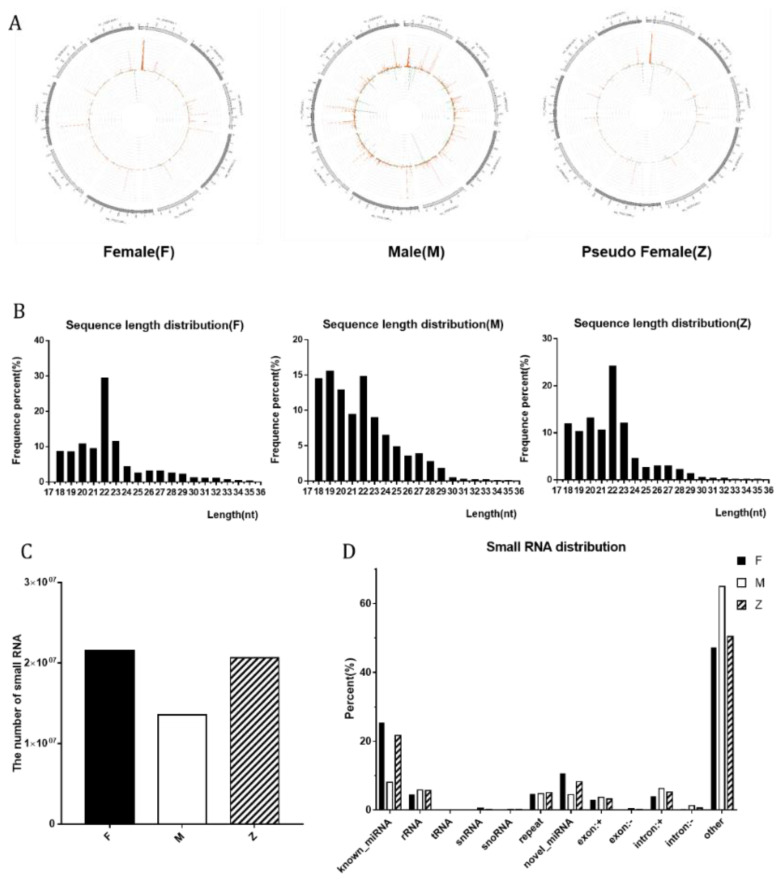
The distribution and number of small RNA among males, females, and pseudo-females: (**A**) Distribution of small RNAs of female, male, and pseudo-female individuals on the genome. (**B**) The sequence length distribution of small RNAs of female, male, and pseudo-female individuals. (**C**) The number of small RNAs of female, male, and pseudo-female individuals. (**D**) The small RNAs distribution of female, male, and pseudo-female individuals.

**Figure 3 genes-12-01696-f003:**
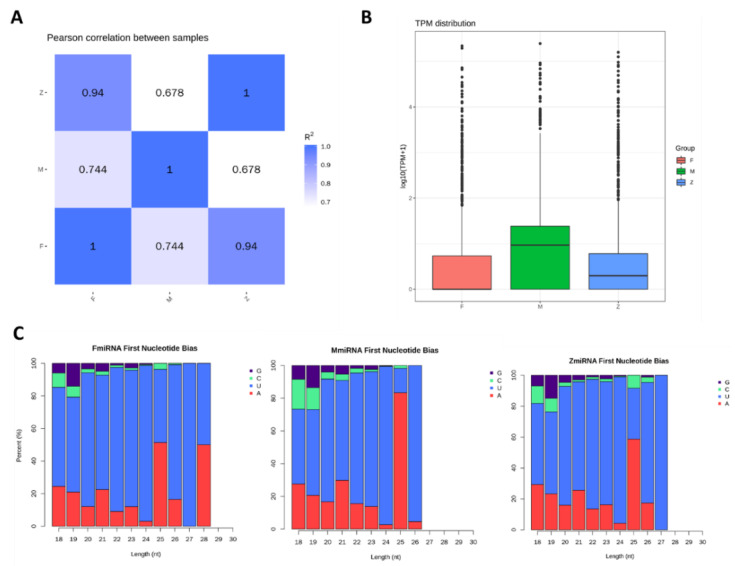
The microRNAs identified by males, females, and pseudo-females: (**A**) Correlation of miRNAs expression levels between samples. (**B**) The TPM distribution of males, females, and pseudo-females (F, M, Z). (**C**) The miRNAs first nucleotide bias of males, females, and pseudo-females (F, M, Z).

**Figure 4 genes-12-01696-f004:**
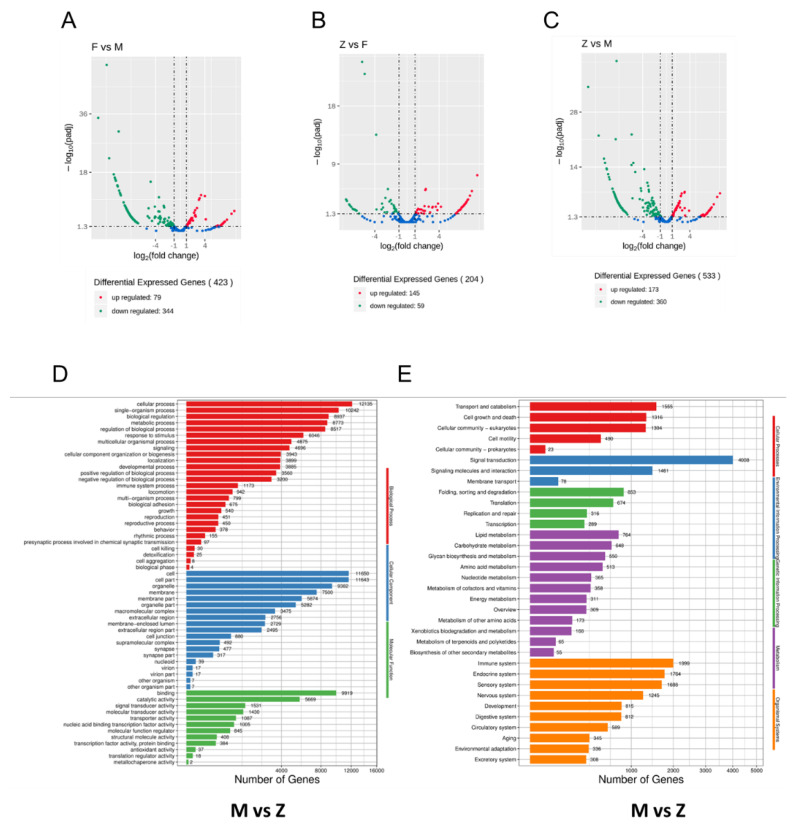
Differential miRNAs analysis of males, females, and pseudo-females (F, M and Z): (**A**) The differential expressed miRNAs between female and male individuals. (**B**) The differential expressed miRNAs between female and pseudo-female individuals. (**C**) The differential expressed miRNAs between male and pseudo-female individuals. (**D**) GO enrichment analysis of target genes of differential miRNAs between male and pseudo-female individuals. (**E**) KEGG enrichment analysis of target genes of differential miRNAs between male and pseudo-female individuals.

**Figure 5 genes-12-01696-f005:**
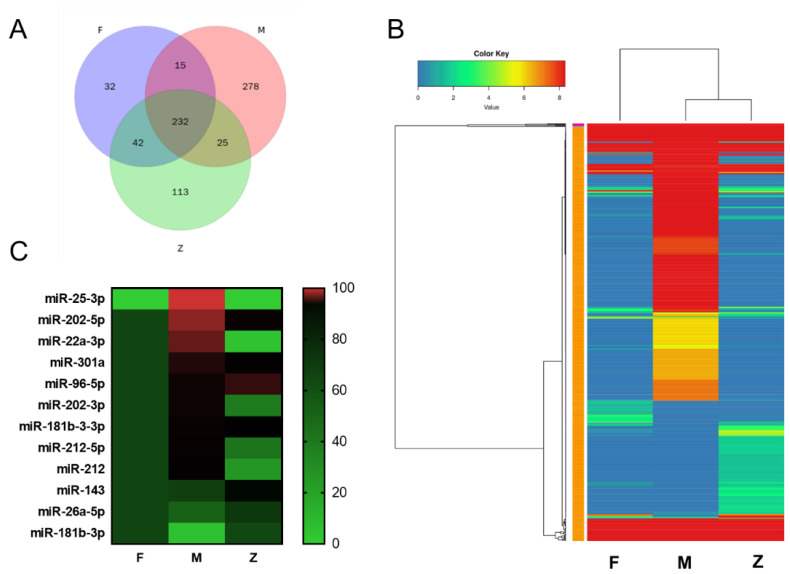
Cluster analysis of differential miRNAs of males, females, and pseudo-females: (**A**) Wayne analysis on unique miRNAs identified in three groups. (**B**) The cluster map of differential miRNAs expression. F: female, M: male, Z: pseudo-female. (**C**) The cluster map of differential miRNAs expression related to gonadal development. F: female, M: male, Z: pseudo-female.

**Figure 6 genes-12-01696-f006:**
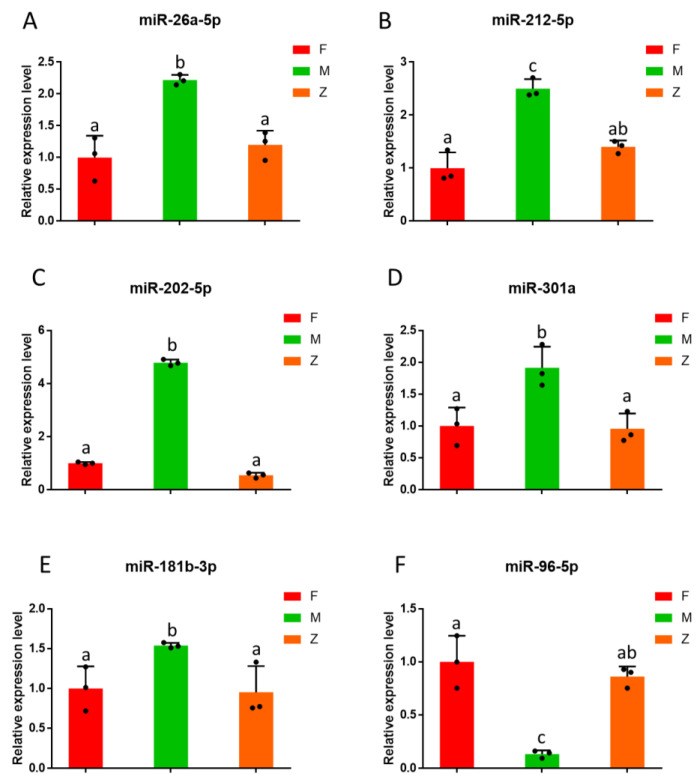
The relative expression level of differential miRNAs among males, females, and pseudo-females: (**A**) The relative expression level of miR-26a-5p among males, females, and pseudo-females. (**B**) The relative expression level of miR-212-5p among males, females, and pseudo-females. (**C**) The relative expression level of miR-202-5p among males, females, and pseudo-females. (**D**) The relative expression level of miR-301a among males, females, and pseudo-females. (**E**) The relative expression level of miR-181b-3p among males, females, and pseudo-females. (**F**) The relative expression level of miR-96-5p among males, females, and pseudo-females. Data were expressed as mean ± SD. Different letters show significant differences (*p* < 0.5).

**Figure 7 genes-12-01696-f007:**
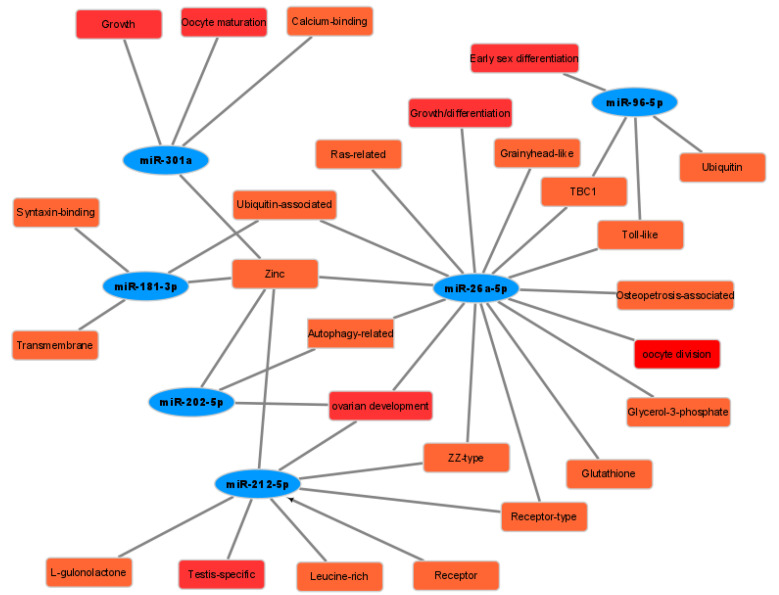
The network of some miRNAs and target genes related to gonadal development. Blue ellipse represents miRNAs differential expressed among males, females, and pseudo-females. Red rectangle represents target genes related to gonadal development differential expressed among males, females, and pseudo-females. Orange rectangle represents target genes differential expressed among males, females, and pseudo-females.

**Table 1 genes-12-01696-t001:** The primers used for amplification of sex specific markers and RT-qPCR of selected miRNAs.

ID	Primer Sequence (5′-3′)	
miR-26a-5p	Forward	GCGCGCTTCAAGTAATCCAGGA
	Reverse	GCAGGGTCCGAGGTATTC
	RT^1^	GTCGTATCCAGTGCAGGGTCCGAGGTATTCGCACTGGATACGACAGCCTA
miR-212-5p	Forward	GCGCACCTTGGCTCTAGACTG
	Reverse	GCAGGGTCCGAGGTATTC
	RT^1^	GTCGTATCCAGTGCAGGGTCCGAGGTATTCGCACTGGATACGACAGTAAG
miR-202-3p	Forward	GCGCGCAGAGGTGTAGAGCATG
	Reverse	GCAGGGTCCGAGGTATTC
	RT^1^	GTCGTATCCAGTGCAGGGTCCGAGGTATTCGCACTGGATACGACTTTTCC
miR-301a	Forward	GCGCGCCAGTGCAATAGTATTG
	Reverse	GCAGGGTCCGAGGTATTC
	RT^1^	GTCGTATCCAGTGCAGGGTCCGAGGTATTCGCACTGGATACGACCTTTGA
miR-202-5p	Forward	GCGCGCTTCCTATGCATATACC
	Reverse	GCAGGGTCCGAGGTATTC
	RT^1^	GTCGTATCCAGTGCAGGGTCCGAGGTATTCGCACTGGATACGACCAAAGA
miR-96-5p	Forward	GCGCGCTTTGGCACTAGCACATT
	Reverse	GCAGGGTCCGAGGTATTC
	RT^1^	GTCGTATCCAGTGCAGGGTCCGAGGTATTCGCACTGGATACGACAGCAAA
miR-181b	Forward	GCGCGCCTCACTGATCAATGAA
	Reverse	GCAGGGTCCGAGGTATTC
	RT^1^	GTCGTATCCAGTGCAGGGTCCGAGGTATTCGCACTGGATACGACTTTGCA
U6	Forward	CTCGCTTCGGCAGCACA
	Reverse	AACGCTTCACGAATTTGCGT
	RT^1^	GTCGTATCCAGTGCGTGTCGTGGAGTCGGCAATTGCACTGGATACGACACTGCTG
4085	Forward	GTTTGAAGTGCTGCTGGGAAG
	Reverse	TTCCCCGTATAAAGCCAGGG
actin	Forward	GTGTATGCAACTCTTCCCTCTCCTATTC
	Reverse	AGCTTCCATTCGGTCTTGTCCTG

RT^1^: reverse transcription.

**Table 2 genes-12-01696-t002:** Data output statistics of three samples (F, M, Z).

Exported Data						
Sample	Reads	Bases	Error_rate	Q20	Q30	GC_content
F	27,912,069	1.396 G	0%	100%	99%	49%
M	24,060,307	1.203 G	0%	100%	99%	51%
Z	31,682,467	1.584 G	0%	100%	99%	50%

**Table 3 genes-12-01696-t003:** The small RNA data statistics of three samples (F, M, Z).

Sample	Total_Reads	Total_Bases	Uniq_Reads	Uniq_Bases
F	22,867,409	511,199,551	374,086	9,254,425
M	14,375,684	311,711,063	428,327	10,622,778
Z	21,932,331	477,377,965	462,946	11,181,717

**Table 4 genes-12-01696-t004:** The expression level of differential miRNAs related to gonadal development.

miRNA	F	M	Z
miR-143	66,453.90	69,180.60	96,007.02
miR-26a-5p	70,778.96	55,517.76	77,512.06
miR-22a-3p	2.20	111.12	0.16
miR-212-5p	13.97	62.99	8.98
miR-181b-3-3p	8.28	44.57	16.66
miR-212	10.22	25.55	4.08
miR-202-3p	3.23	24.36	1.96
miR-301a	1.04	18.42	2.61
miR-202-5p	0.13	8.91	0.65
miR-25-3p	0.00	8.32	0.00
miR-96-5p	68.58	7.72	25.81
miR-181b-3p	6.21	0.59	6.04

## Data Availability

Data available in a publicly accessible repository.

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
