# Peer review of "MicroRNAs May Play an Important Role in Sexual Reversal Process of Chinese Soft-Shelled Turtle, Pelodiscus sinensis"

_genes, 2021, doi:10.3390/genes12111696_

Round 1

Reviewer 1 Report

Revision for the authors, manuscript ID “genes-1409414”:

Lines 50-51: What kind of medicine value does Pelodiscus sinensis has? What about the nutritional value? Until further knowledge about that topic is collected, statements about “nutritional and medical values” must be removed. You must write that this species is highly exploited in traditional Chinese medicine (TCM), but that this practise of exaggerated wildlife exploitation (including the one coming from TCM) is one of the main causes of biodiversity loss. Biodiversity indeed represent the most important resource for humankind. Cite https://doi.org/10.1371/journal.pone.0246081 and https://doi.org/10.2744/CCB-0747.1 .

Line 61: 22th instead of 22th. Thanks

Line 63: genes names must be written in italics (e.g., Sox9). Correct throughout the ms if needed.

Line 72: Check spaces. Thanks

Line 75/97: (i.e., female, male, and pseudo-female). Thanks

Line 79: “aquaculture sector that should be better regulated in order to reduce the risk associated with Chinese soft-shelled turtles breeding. Cite https://doi.org/10.1007/s00114-018-1558-9 . Thanks

Line 61 and 84: Be coherent throughout the text. Check which is the correct form; either 30°C or 30 °C.

Line 85: Provide the name and brand of the commercial feed. Thanks

Line 89: Add space after 10. Thanks

General: Before all citations (e.g., [74]), you must put a space. Please be coherent throughout the text.

Line 99: You do not specify if they did DNase treatment or not. The same for rRNA removal. Please clarify in detail what you did. Thanks

Line 105: There is no such thing as a "miRNA library". This is a small RNA library. Please clarify.

Line 107: Remove the full-stop before USA. Thanks

Line 108: Correct “end” in “ends”

Line 112: Are these scripts available? Insert them in supplementary material. Thanks

Lines 114-115: These lines are not clear at all. Q20/30 were calculated and then? If the reads were not filtered from quality using these calculations how are the analysis based on high quality clean data? In addition, PRINSEQ was used to remove low quality reads, meaning? Below which quality threshold? and why not trimming the reads? Please clarify all these points in detail providing the missing information. Thanks

Line 121: Which genes? What database? This is not mentioned anywhere in the methods section. These points must be well-clarified and specified in the text. Thanks

Line 132: Missing citation.

Line 134: Using only one housekeeping gene to normalize miRNA expression could be not enough. How did you selected U6 rRNA and why you used only one gene? Add citations about the use of U6 as gene for normalization.  See https://doi.org/10.1016/S0003-2697(02)00311-1

Line 136: “were” instead of “was”. Thanks

Line 154: Please provide at least a %. Which was the % of pseudo females in the injected group? Is there a control group of non-injected individuals? You need it to determine the % of pseudo females (male genotype and female phenotype) in non-injected control group.

Line 159: This line refers to a figure and that is fine, but…What about tail measures (mean +- SD) in males, females and pseudo females? Have been these measures taken? If yes provide the values and the number of individuals on which has been done. Thanks

Line 161: Same of line 159. What is it “normal”? What is “slightly smaller”? Provide at least means +- SD. Did you conducted histological investigations to compare ovaries of females and pseudo females? Obviously measured referring to animals of that specific age (also provide mean age of the individuals). Thanks

Lines 159-161: Please provide a table or put the data between brackets along the text. Thanks

Line 166: Please change “F:female,M:male,Z:pseudo female” in “F: female, M: male, Z: pseudo female.” Thanks

Line 167: Strange comma in the caption. Substitute this symbol with “,”. Same for the other analogous situations (e.g., lines 175, 177, 201). Thanks

Line 179: pseudo not capitalized. Thanks

Lines 186-188: Put spaces after “nt” and after “%”. Thanks

Line 189: Put the reference accession number of turtle genome and eventually the proper citation. Thanks

Line190: Add a comma after 20669068. Thanks

Line 192: “than the one of pseudo…”

Line 193: “contributed an important role” is wrong in English. Please re-phrase with a native speaker. Thanks

Line 195: miRNAs instead of miRNA.

Line 209: Please define “genetic genders”. Should not genetic genders be only two? Thanks

Line 212: “showed that” and “the” before female. Be coherent throughout the text: use either pseudo females or pseudo-females. Choose one. Thanks

Line 264: “Differently from males and females,”…

Lines 268-270: miRNAs

Line 269: What does it mean “included females and pseudo-females”? Is it not obvious that the others are those two? Please remove.

Lines 274-275: Please put normal commas, not that symbol.

Line 286: Explain the values reported in table 4. What do these numbers represent?

Line 379: This study must now be conducted also on individuals that are not injected unless you do not know if what you are observing derives from the treatment. This should have been done in parallel.

Reviewer 2 Report

This manuscript, entitled, “MicroRNAs played a crucial role on sexual reversal process of Chinese soft-shelled turtle, Pelodiscus sinensis”, by Zhou et al, describes differential profile of miRNA expression in male, female and pseudofemale Chinise soft-shelled turtle. There are serious flaws that prevent the manuscript from being published. 

1. The aim of the study is not precisely formulated. Though the authors underline the higher economical and nutritional value of male individuals, they focus on male to female sex reversal. 

2.  If the authors were interested in sex determination or sex reversal process it is not clear why they pulled together RNA isolated from 5 different tissues and did not focused on gonadal tissue. 

3. Involvement of miRNA in gonadal development has been described before. Therefore authors should clearly point on novelty of their study and they don't do it. 

4. Figures 2, 3, 4, 7 are unreadable.

Minor Comments: 
1. Are there 3 genetic sexes? I would say that there are 2 genetic sexes (pseudofemales are genetic males). 
2. In Results sections subsections’ titles should be more informative (they should inform about main result obtained in given subsection).
3. I suggest to use abbreviation “miRNA” instead of full name “microRNA” consequently. 
4. Avoid spaces before commas. 
5. Descriptions of figures are not comprehensive e.g. Figure 1 
L 281. "crucial" is overestimated. These results suggest that the miRNAs may play role in the process of sexual reversal.
L273. Should be Table 4 instead of table 3

Round 2

Reviewer 1 Report

Dear authors,

please be sure that all the corrections will be completed before publication.

Sincerely, Reviewer 1

Reviewer 2 Report

Though I don't feel qualified to judge about the English language and style I have an impression that extensive editing of English language and style is required.

The authors have improved the manuscript but still I have some comments

Abstract:

L10 -11 “transformation from male to female phenotype” would be more precise

L13 “male, female and pseudo-female” would be more informative than “three types”

L13 add “understanding” after “for”

L14-16 The sentence “Injecting …” is not needed in the abstract. It is a detail appropriate for intro.

L 21 “upregulated” instead of “upregulate” 2x

L 22 Why the total number of 602 differential miRNA is bigger than 533 differentially expressed miRNA? It is not clear for a reader.

L24 “their” instead of "the”

L25-27 Instead of the sentence I would suggest: The TR-qPCR results confirmed differential expression of 6 of 12 miRNA: miR-26a-5p ……..

L27 I suggest “Using these 6 miRNA and some of their target genes we constructed a network diagram related to gonadal development” 

L28 “We suggest” instead of “Therefore”

Introduction:

L23-48 I would remove this part of intro. It is obvious information for “Genes” readers.

L49-87 There is no flow in this part of Introduction. I suggest following flow:

  1. Sexual dimorphism in the species.
  2. Features of sexual dimorphism
  3. Advantages of males
  4. GSD (ZZ/ZW)
  5. Sex reversal using E2
  6. Molecular mechanism of sex reversal in unknown
  7. Part about miRNA (what do you mean writing “in the present review”???? L70)
  8. Aim of the study

L84 “to” instead of “which”

Materials and methods/Discussion

In the first round of comments I wrote “2.  If the authors were interested in sex determination or sex reversal process it is not clear why they pulled together RNA isolated from 5 different tissues and did not focused on gonadal tissue.” I accept the answers given me but I expect that the clarification should be also within the manuscript.  

Results:

L105 what does “sex-specific markers” mean? Are they sequences specific for W chromosome???

L108 Instead of the complicated sentence I would write “Animals were divided into three groups: males, females and pseudo-females”.

How many animals were in each group?

L109 “animal” instead of “group”

L191 “phenotypes” instead of “three types”

L192 Improve the Table description. The primers used for amplification of sex specific markers and RT-qPCR of selected miRNA???

L194 and everywhere “males, females and pseudo-females” instead of “three types”

L195-199 This part is repetion from subunit 2.3 and 3.1

L200 It is conversational/colloquial style

L232 Use “miRNA” instead of “microRNA”

Table 4. Use “miRNA” instead of “sRNA”  

Conclusions

In this subsection only the sentence “6 miRNA such as …….” fits to  conclusion section. The rest of text doesn’t represent conclusions.

In the whole manuscript

Avoid double space!!!

You use sometimes small RNA and sometimes sRNA. Decide for one form.

Figures 2, 3, 4, 7 are still unreadable (especially Fig. 2A) – not acceptable
